# Economy and elderly population, complementary or contradictory: A cross-continental wavelet coherence and cross-country Granger causality study

**Kethaka Galappaththi**[1], **Ruwan Jayathilaka**[1]\*, **Lochana Rajamanthri**[1], **Thaveesha Jayawardhana**[2], **Sachini Anuththara**[2], **Thamasha Nimnadi**[2], **Ridhmi Karadanaarachchi**[2]

1 Department of Information Management, SLIIT Business School, Sri Lanka Institute of Information Technology, Malabe, Sri Lanka, 2 SLIIT Business School, Sri Lanka Institute of Information Technology, Malabe, Sri Lanka

\* ruwan.j@sliit.lk

## Abstract

The aim of this study is to explore the causal relationship between the economy and the elderly population globally as well as continent-wise. This research was designed as a continent-wide study to investigate the differences between several regions simultaneously. The economy was measured by the Gross Domestic Product (GDP) per capita growth rate while the population aged above 65 as a percentage of the total was considered the elderly population. A panel dataset published by the World Bank for a period of six decades from 1961 to 2020 covering 84 countries was used as data for the analysis. Wavelet coherence was the methodology used for the study since it was considered suitable to present causality as well as the causal direction between the two variables for different sections during the six decades. Thereafter, Granger causality was applied for a cross-country analysis to gain further insights on the causality of individual countries over the years. Findings of the study reveal that the causality and its direction have been changing over time for most continents. Negative correlations with the leading variable interchanging with time are evident for the majority of the regions. Nevertheless, results indicate that in a global perspective, elderly population predominantly leads the economic growth with a positive correlation. Research approach allows ascertaining the short-term and medium-term changes that occurred concerning the direction of the relationship throughout the stipulated period of the study, which could not be drawn by any previous study. Even though region-wise literature is available on this topic, global studies for decades have not been conducted yet.

## Introduction

The elderly population has been a decade-long burning issue that has received close attention from individual nations and international institutions for years since the United Nation (UN)

**Data Availability Statement:** All relevant data are within the manuscript and its with Supporting information files.

**Funding:** The authors received no specific funding for this work.

**Competing interests:** The authors have declared that no competing interests exist.

General Assembly convened the first World Assembly on Ageing in 1982. Even though the elderly population is not a new aspect throughout world history, the focus on this age cohort has been growing over the past decades due to the surge in the elderly population as a proportion. According to the World Development Indicators of the World Bank [1], 6% of the elderly population in the 1980s has grown to 7% in the 1990s, and 10% in 2021. In general, an increase in life expectancy with medical advancements and low fertility rates have been identified as the root causes for the rapid growth of the elderly population. However, it is imperative to reflect on the consequences of the situation and to set national policies accordingly. Studies related to the influence directed by the elderly population on various macroeconomic factors have already been conducted. Major variables considered to investigate the relationship and causality with the elderly population are unemployment, inflation, government expenditure, and health costs [1–6]. Whilst a considerable amount of research has been conducted about the elderly population, research community is yet to understand its overall consequences. Another significant macroeconomic factor to be concerned about is economic growth.

The main aim of the study is to explore the nature of the relationship between the economy and the elderly population across an extended period with variations during different periods of the given years. This research was designed as a continent-wide study to simultaneously investigate the differences between several regions. Furthermore, a cross-country analysis was conducted using Granger causality to understand the relationship between the economy and the elderly population among individual countries. It is because countries have implemented diverse policies and varying macro-economic conditions exist, which would impact on the relationship of the two variables within a given country. The significance of this study is evident in many different aspects. Firstly, the novelty of the research is due to the initial usage of the wavelet coherence approach, as the methodology adopted to investigate the nature of the relationship and causality between the economy and the elderly population. This approach allows ascertaining the short-term and medium-term changes that occurred concerning the direction of the relationship throughout the stipulated period of the study, which could not be drawn by any previous study. Secondly, the study considers over 80 countries worldwide representing all habitable continents for 60 years. Even though region-wise literature is available on this topic, global studies for decades have not been conducted yet. Thirdly, the ability of this research to distinguish variations within the stipulated period for each region would be helpful. This will assist to ensure the effectiveness of previous policy implementations related to the economy and the elderly population, and its learnings could be used for future decision making.

The rest of the paper has been structured as follows. The following section presents the literature review. The section thereafter presents the material and methods used in the study. The fourth section describes the results obtained by analysing the data using the wavelet coherence approach along with the discussion of the results. Finally, conclusions drawn from the study are given in section five.

## Literature review

Opinion of research community is divided regarding the relationship between economic growth and the elderly population. A substantial research studies conclude a negative relationship [7–12] while a considerable amount of existing literature suggest a positive relationship [13–16]. Despite the fact that the research community has shown interest about the nature of the relationship and causality between the elderly population and economic growth, the importance of identifying its relationship is unchallenged. Even with the ample body of literature available in this field, the change in the nature of the relationship between the economy

and the elderly population is yet to be investigated in the global scale for a prolonged period. At present, a few regional studies have been conducted across regions [17, 18] since individual countries [19–22] had been the main focus in numerous research studies. The existing literature had been using different methodologies such as regression [23, 24] and granger causality [25–27] to identify the effects one variable has on the other and the direction of those two variables. However, so far, according to available information and researcher's knowledge, no study has ever been conducted to ascertain the relationship between the economy and the elderly population across habitable continents for several decades, with the emphasis given for changes to the nature of the relationship along that time period. Yet, it is of utmost importance to look into such changes concerning the relationship. It is because national, regional, and international institutions have implemented various policies from time to time, which would have affected the nature of this relationship. Thus, a clear literature gap is visible which is to be filled from the findings of this research study.

When considering the overall global impact of the aging population, mixed opinions can be seen. The world population is aging quickly, and certain parts of the world are experiencing decreased fertility rates [28]. As a result, countries with low fertility rates are isolated from the world's industrial sector, which is inherently labour-scarce and could worsen the issue of aging population in the future. However, according to the United Nations [29], nations like China and India have steady fertility rates and will manage labour shortages well in the future. Considering Gross Domestic Product (GDP) growth and capital stock as dependent variables, Mahmoudinia, Kondelaji [30] employed panel cointegration and causality techniques to demonstrate that there is a long-run link between the said variables. Over time, population has a favourable and statistically significant effect on economic growth.

Although sufficient literature is available about the relationship between the elderly population and economic growth, these lack the insights from different periods of time as it is a known fact that the relationship would change due to short-term and medium-term policy implementations and other macro-economic conditions. Thus, the model derived in this study fills the literature gap that is yet to be filled.

## Materials and methods

### Data

The two variables, economic growth and the elderly population investigated in this study were quantified from two economic indicators. The annual GDP per capita growth rate was used to measure the economic growth, while the population aged 65 and above as a percentage of the total was considered as the elderly population. Secondary data available in the World Bank Indicators online database were used for the study. Yearly figures for the above-mentioned two economic indicators were taken from 84 countries (27 from Africa, 15 from Asia, 14 from Europe, 12 from North America, 3 from Oceania, and 13 from South America) over six decades from 1961 to 2020. Altogether 5,040 observations were collected and analysed for the research. Descriptive statistics of the dataset is provided in the S1 Appendix.

### Methodology

Wavelet coherence analysis was used to analyse the causality of two variables along a time series. It was popularised by Christopher Torrence, who initially applied it for meteorology in 1998 [31]. Over time, it has been applied to diverse fields, from Economics [32] to Medicine [33]. Several research studies have further contributed to the development of the model, including the wavelet transformation of the two time series and the wavelet coherence at difference phases [34–37].

Real-world data consist of subtle oscillations that could be important for gaining insights. Although Fourier analysis could be used to represent certain trends, it is inefficient in capturing abrupt changes. A wavelet is a rapidly decaying wave-like oscillation that has zero mean and exists for a finite duration and in different sizes and shapes [38]. Continuous and discrete are the two main wavelet transforms. Those transformations occur based on the way wavelets are shifted and scaled. A wavelet, which was constructed by the function $\psi^{a,b}(x)$ with contraction, $a$ and translation, $b$ can be mathematically denoted as follows,

$$\psi^{a,b}(x) = |a|^{-\frac{1}{2}}\psi\left(\frac{x-b}{a}\right) \qquad (1)$$

Wavelet coherence can even be considered as an extension of the wavelet analysis. In bivariate analysis, the wavelet coherence approach can explain how one variable leads the other. The superiority of wavelet coherence over conventional techniques such as Granger causality is that it can explain the intensity and direction of the causality between two variables along the period under consideration. In wavelet coherence, with the analysis of two time series, details of which variable leads the other variable could be determined along with its direction. The wavelet coherence formula [39], including smoothing factor, $s$, first variable, $y$ and second variable $x$, is as follows,

$$Coherence = \frac{|sWave.xy|^2}{sPower.x \cdot sPower.y} \qquad (2)$$

In this study, the economic growth measured from an annual GDP per capita growth rate was considered as the first variable, while the elderly population is taken as the population aged 65 and above as a percentage of the total was considered as the second variable. Data file used for the study is presented in S2 Appendix and the Wavelet coherence diagrams were created using R software.

Granger causality test was applied for individual countries to supplement and further enhance the results obtained from wavelet coherence analysis. Considering X and Y, two stationary covariance variables were tracked over T periods and N cross-section units. Granger [40] defines causality for each individual $i$ [1,N] as follows: the variable $X_{i,t}$ causes $Y_{i,t}$ if we are better able to predict $Y_{i,t}$ by using all available information than if the information apart from $X_{i,t}$. Consider time stationary VAR representation for a panel data set, assuming the Granger causality model is linear. For each cross-section unit $i$ and time period t, the following Eq (3) is estimated.

$$Y_{i,t} = \sum_{k=1}^{P}\beta_k Y_{i,t-k} + \sum_{k=0}^{P}\theta_k X_{i,t-k} + u_{i,t} \qquad (3)$$

There $u$ is normally distributed with $u_{i,t} = \alpha_i + \varepsilon_{i,t}$, $p$ is the number of lags. It is assumed that the autoregressive coefficients $\beta_k$ and the regression coefficients $\theta_k$'s are constant for $k \in$ [1, N].

Furthermore, as diagnostics tests, impulse response of the per capita GDP and elderly population for all countries was conducted along with root of companion matrix.

## Results and discussion

The primary focus of this study was to determine the causality of the economy and the elderly population with its direction over time in diverse continents. A total of 5,040 observations from a secondary panel dataset were analysed using the wavelet coherence approach capturing

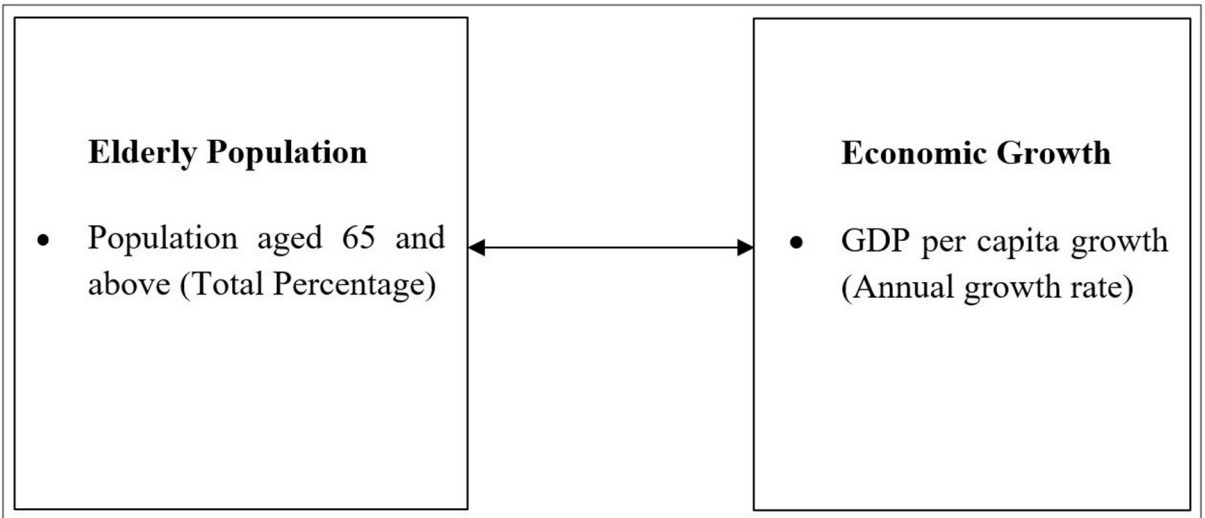

**Fig 1. Theoretical framework of the research study.** Source: Authors' illustration based on past literature.

all continents (Africa, Asia, Europe, Oceania, North America, and South America) excluding Antarctica since it is not considered a habitable continent. Theoretical framework of the research study can be illustrated as follows.

The theoretical framework in Fig 1 clearly depicts the study aim, to identify/understand the nature between two endogenous variables, elderly population and economic growth, operationalised by the population aged 65 and above as a percentage from the total population and annual GDP per capita growth rate, respectively.

Figures below depict the findings in terms of continents considered for the study.

Fig 2 illustrates the movement of mean annual GDP per capita growth rate and the mean elderly population (as a percentage of the total) over time for each continent. Africa, Asia, Europe, North America, Oceania, and South America were denoted by AF, AS, EU, NA, OC, and SA, respectively. The series with the prefix, PGDP, is the movement in the mean annual GDP per capita growth rate and series with the prefix, EPOP, is the movement in the mean elderly population (as a percentage of the total). The primary (leftward) vertical axis represents the values for EPOP while the secondary (rightward) vertical axis represents the values for PGDP. A change in the mean annual GDP per capita for all regions has been mostly positive until 2020 with a few exceptions. This explains the continuous economic development experienced by the regions. Notably, 2020 was an exceptional year because covid-19 global pandemic negatively affected all regional economies. The only period that the GDP per capita growth rate has gone down considerably was 1975, due to the global recession experienced from 1973 to 1975 [41]. The elderly population across the regions have grown steadily over the years. However, it is clear that the growth rate of the elderly population as a proportion is comparatively much higher in the European region than in the rest of the region. Proper elderly care, high living standards, and healthy lifestyles have probably increased the life expectancy of Europeans, in turn, increasing the elderly population proportion [42].

The wavelet coherence graphs for the six continents and all countries generated using R-software have been provided in this section along with interpretation and the discussion. As mentioned earlier, the economic growth and the elderly population were considered as the first and second variables, respectively. The correlation and causality of the two variables can be interpreted using attributes in the graphs.

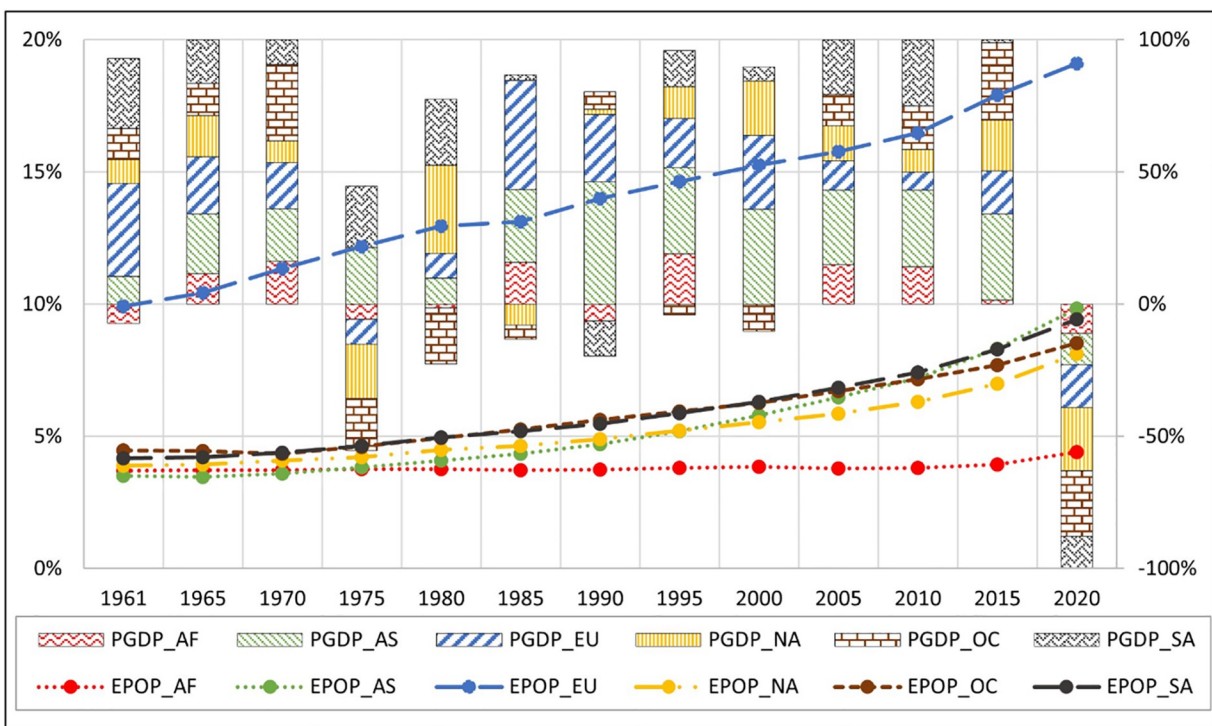

**Fig 2. Combination chart on mean annual GDP per capita growth rate and mean elderly population (as a percentage of the total) over time.** Source: Authors' compilation based on World Development Indicators.

The direction of the arrows in Fig 3 indicates whether the variables move in-phase (rightward arrow indicating a positive correlation), or anti-phase (leftward arrow indicating a negative correlation). The cold (blue) regions of the Figure indicate no correlation while the warm (red) regions depict the analysed variables are correlated.

In Fig 4, all countries between 1960 and 1970, in the medium-term (medium frequency), rightward arrows portray a positive correlation between economic growth and the elderly

| Attributes | Interpretation |
|---|---|
| ↗ | Second variable causes the first variable (In phase) |
| ↘ | First variable causes the second variable (In phase) |
| ↖ | Second variable causes the first variable (Out phase) |
| ↙ | First variable causes the second variable (Out phase) |
| Low frequency | 0.0 - 0.3 |
| Medium frequency | 0.3 - 0.7 |
| High frequency | 0.7 – 1.0 |

**Fig 3. Interpretation of wavelet coherence.** Sources: Authors' compilation.

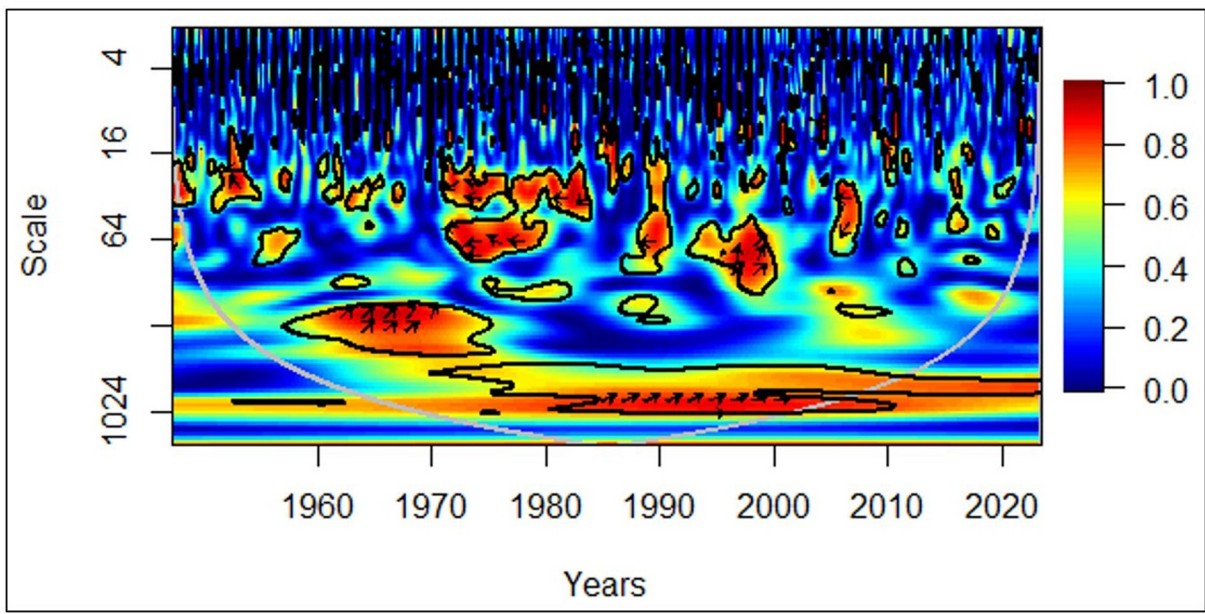

**Fig 4. All countries: GDP vs EPOP.** Source: Authors' illustration.

population. The rightward and upward arrows depict that the elderly population leads to economic growth. Moreover, between 1985 and 2000, in the long-term (low frequency), rightward arrows portray a positive correlation between economic growth and the elderly population. The rightward and upward arrows in the same cluster also depict that the elderly population leads to economic growth.

Interestingly, the effect of the elderly population leading to economic growth was only for the medium-term in the 1960s and 1970s. This pattern was evident in the long-term only after 1985. It is likely the lost lives in World War II have played a huge role in this irregularity. The number of deaths was estimated to be in the range of 62 million (Mn) and 78 Mn during the war in the 1940s [43]. The higher proportion of them were young soldiers, which had made an artificial demographic shift with a less proportion of elders living in the 1970s, depicting a medium-term effect. In the second observation of the elderly population leading to positive economic growth, its intensity has diminished in the last two decades. The possible reason can be the recognition of elderly care in recent years. Currently, with pension plans and subsidies in place in most countries, people tend to retire early [44], limiting their contribution to the economy. Even though the concept of elderly care existed in the 20th century, people tend to work until later stages in their lives contributing to the economy. Comparatively, a few governments have imposed a retirement age at that time.

In Fig 5, between 1978 and 1982, at the medium-term (medium frequency), the arrows facing the left mirror evidence of negative correlations between economic growth and the elderly population. The leftward and downward arrows demonstrate that the elderly population leads to economic growth. Furthermore, between 2008 and 2015, in the medium-term and long-term (medium frequency and low frequency), the arrows facing right show a positive correlation between economic growth and the elderly population. The rightward and upward arrows indicate that the elderly population leads to economic growth.

Renowned as the dark continent in the world, Africa was in bleak conditions in terms of health, education, and many other aspects of quality life till the end of the 20th century.

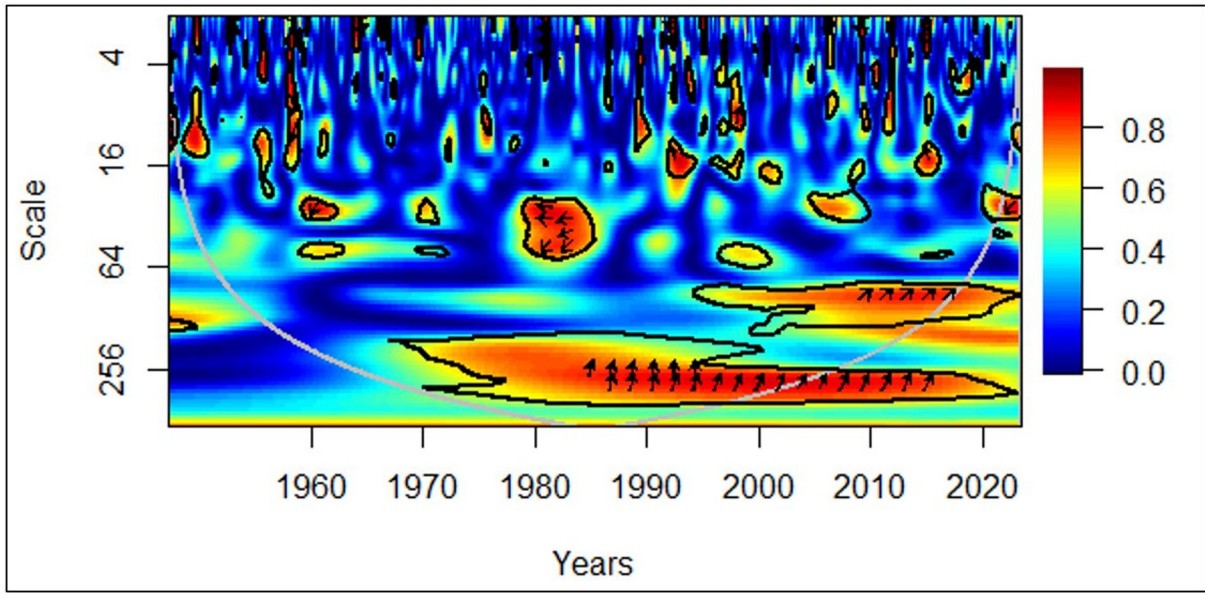

**Fig 5. African region: GDP vs EPOP.** Source: Authors' illustration.

Therefore, more healthy Africans in their old age must have started contributing to the economy relentlessly from the latter part of the 20[th] century. During the mid-20[th] century, it was a rare occurrence for Africans to live more than 65 years. Even then, a majority of them would have been in poor health. Tendering to the aged and its costs must have negatively impacted the economic growth during that period. The tendency for the elderly population to contribute to the economic growth in Africa emerged recently in contrast to the global perspective, which started to decline in the last few years. A major factor for this deviation can be the improving life expectancy of Africans [45]. Occasional leftward and downward arrows, that depict the elderly population leading to adverse economic growth were validated by existing studies.

In Fig 6, in 1960s, in the medium-term (medium frequency), rightward arrows depict evidence of the positive correlation between economic growth and the elderly population. The rightward and upward arrows demonstrate that the elderly population leads to economic growth. Between years 2000 and 2010 in the short-term (high frequency), leftward arrows depict evidence of the negative correlation between economic growth and the elderly population. The leftward and upward arrows portray economic growth lead by the elderly population.

Since most Asian countries were in the developing stage during the 1960s, its governments were unable to provide sufficient financial assistance for the aged compared to similar policies adopted in the countries in the European and North American continents [46]. Therefore, the elderly should have engaged in economic activities to meet their basic needs, in turn, contributing to economic growth. Additionally, high regard for the elderly in Asian countries might have resulted in them actively participating in the economy. The existing literature by the UN confirms the positive contribution as discussed in the literature review. It is surprising to see that the economic growth negatively affected the elderly population in the 2000s since the common belief is that the economic growth would raise living standards increasing life expectancy. However, unhealthy diets and the lack of physical activity attached to busy and stressful lifestyles, which are by-products of economic growth have resulted in premature deaths from

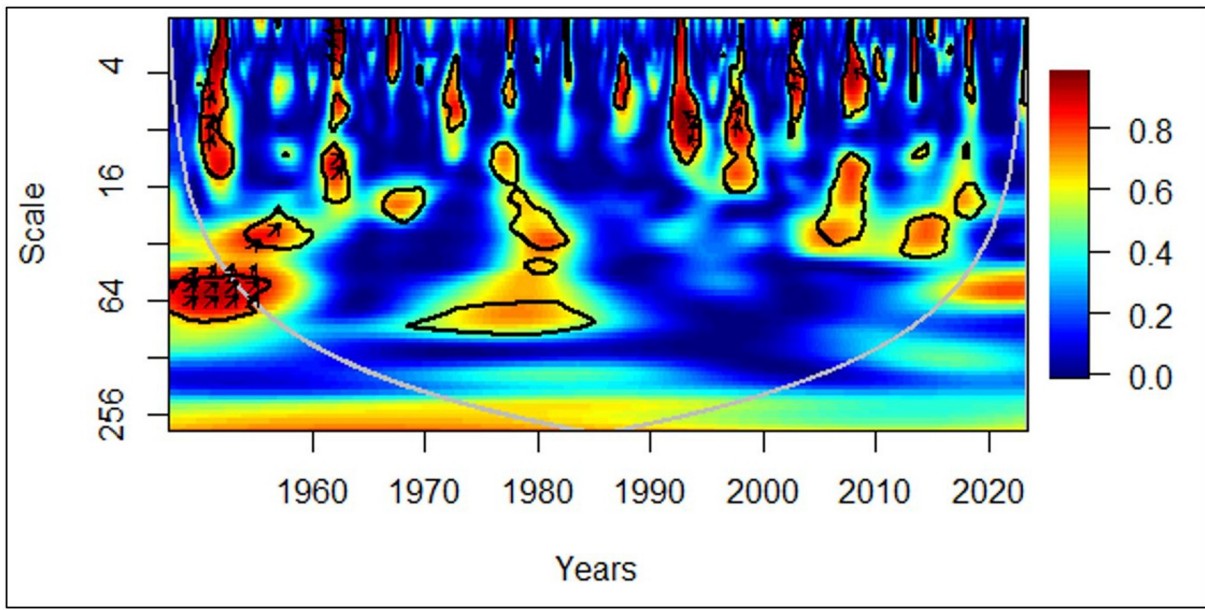

**Fig 6. Asian region: GDP vs EPOP.** Source: Authors' illustration.

non-communicable diseases in Asian regions between years 2000 and 2010 [47]. Nevertheless, it is shown as short-term because governments and health institutes have intervened to tackle this issue afterwards [48].

In Fig 7, between 1960 and 1970, the arrows facing left in the medium-term (medium frequency) denote a negative correlation between economic growth and the elderly population. Additionally, leftward and downward arrows depict that the elderly population leads to

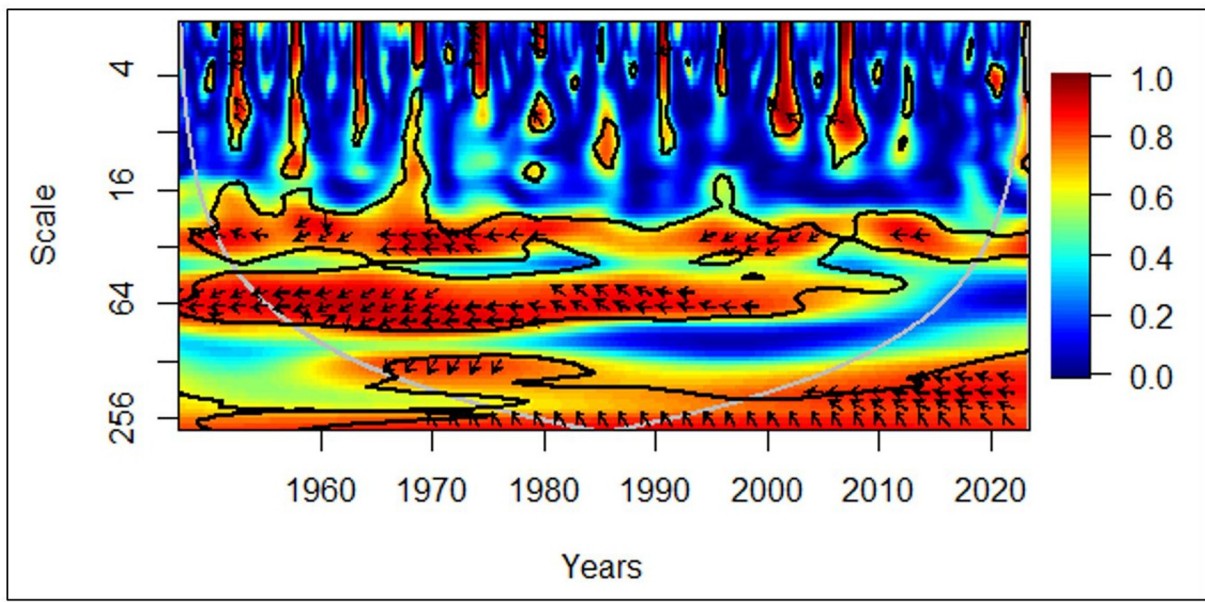

**Fig 7. European region: GDP vs EPOP.** Source: Authors' illustration.

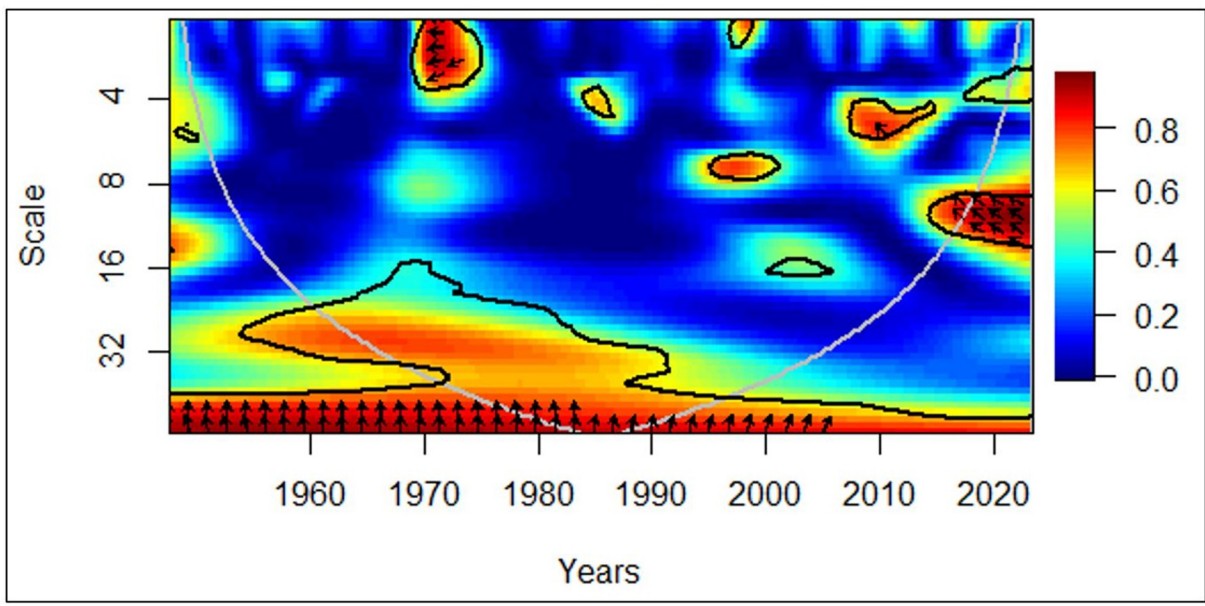

**Fig 8. Oceania region: GDP vs EPOP.** Source: Authors' illustration.

economic growth. Moreover, between 1980 and 1990, the arrows facing left in the medium-term (medium frequency) show a negative correlation between economic growth and the elderly population. The leftward and upward arrows depict that economic growth leads to the elderly population.

Due to the comparatively higher quality of life experienced in the European region and more focus being given to the elderly even in the mid-20th century, older persons were cared for regardless of them contributing significantly to the economy. Since a considerable proportion of finances was dedicated to the elderly care in Europe [49], economic growth has been negatively impacted. A research report published by the Austrian Academy of Sciences (OeAW) discussed under the literature review further confirms this finding. Towards the latter part of the 20th century, with the booming economies, the European elderly had to face a different challenge, which was isolation from their communities. As the healthy and young Europeans were busy contributing to the economy, older persons were neglected, exposing them to cognitive disorders such as dementia [50]. It must have been a leading cause of decreasing elderly population along with the economic development.

In Fig 8, at short term (high frequencies) between years 1969 and 1973, the arrows facing left illustrates a negative correlation between economic growth and the elderly population. The leftward and downward arrows illustrate that the elderly population leads to economic growth. Further, between years 2015 and 2020, in the medium-term (medium frequencies), the arrows continue to face leftward illustrating a negative correlation between economic growth and the elderly population. But this time, the arrows are pointed leftward and up representing economic growth leading the elderly population. The same arrow behaviour is visible in the year 2010 as well.

In Oceania, specifically, Australia, a high rate of migration exists due to the booming economies [51] in the 21st century. With young immigrants adding to the regional population, the proportion of the elderly population must have decreased.

In Fig 9, in the medium-term (medium frequency), between years 1990 and 2000, the arrows facing right illustrate a positive correlation between economic growth and the elderly

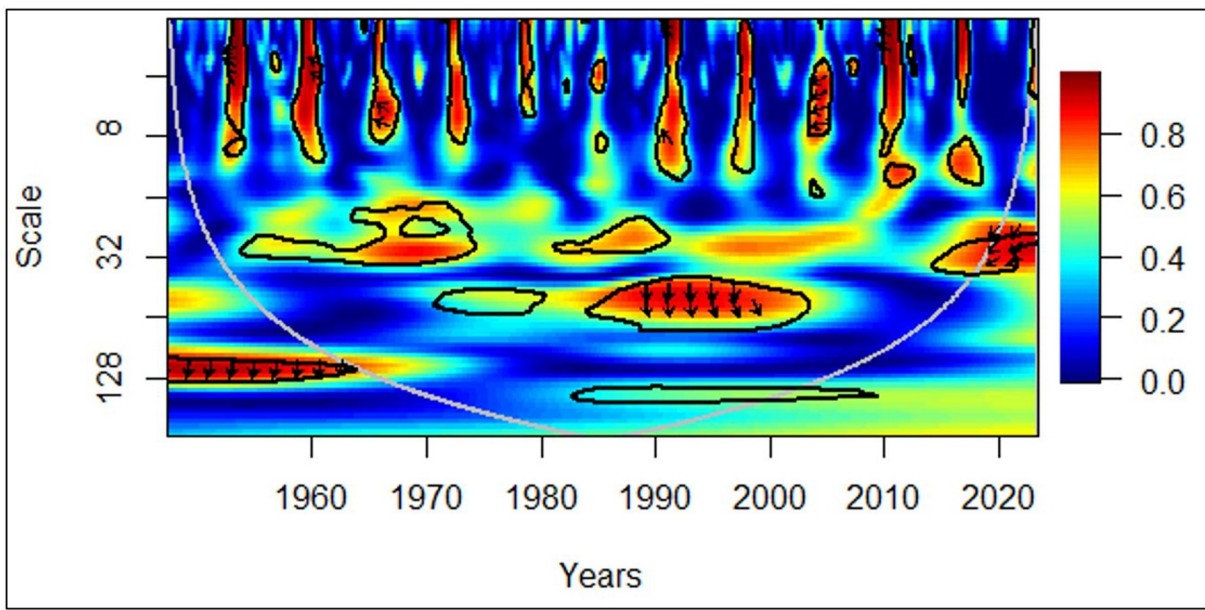

**Fig 9. North American region: GDP vs EPOP.** Source: Authors' illustration.

population. The rightward and downward arrows illustrate that economic growth leads the elderly population. In addition, in 2004, in the short-term (high frequency), the arrows facing left describe a negative correlation between economic growth and the elderly population. The leftward and downward arrows show that the elderly population leads to economic growth.

The rise in living standards and improved elderly care resulting from economic growth in the 1990s should have led to the growth in life expectancy of an average person [52]. Hence, in the medium-term, the elderly population has grown as a percentage of the total population. The tendency to bear high expenditure by the governments for the elderly due to the large proportion of elders in the North American region has had a negative impact on economic growth. This was indicated from the wavelet coherence graph in 2004, and confirmed by the existing literature [53]. Apart from this, a working paper from RAND corporation has revealed a similar finding that was previously discussed in the literature review section.

In Fig 10, between 1980 and 1990, both in the short and medium-terms (medium and high frequencies), leftward arrows portray a negative correlation between economic growth and the elderly population. The leftward and upward arrows depict that economic growth leads to the elderly population. Similarly, in 2006 and 2010, in the short-term (high frequencies), leftward arrows portray a negative correlation between economic growth and the elderly population, while leftward and upward arrows depict that economic growth leads to the elderly population.

In South America, with the economic development, it has been found that the first-birth rates have increased [54]. Therefore, the decline in elderly population as a proportion can be explained by higher birth rates associated with economic growth.

In order to incorporate the institutional, geographic, and cultural differences across the countries, a cross-country analysis was conducted. The reason is because there is a possibility that some countries included in the dataset might have a higher elderly adult population due to better healthcare infrastructure or other institutional differences which might affect GDP differently than others.

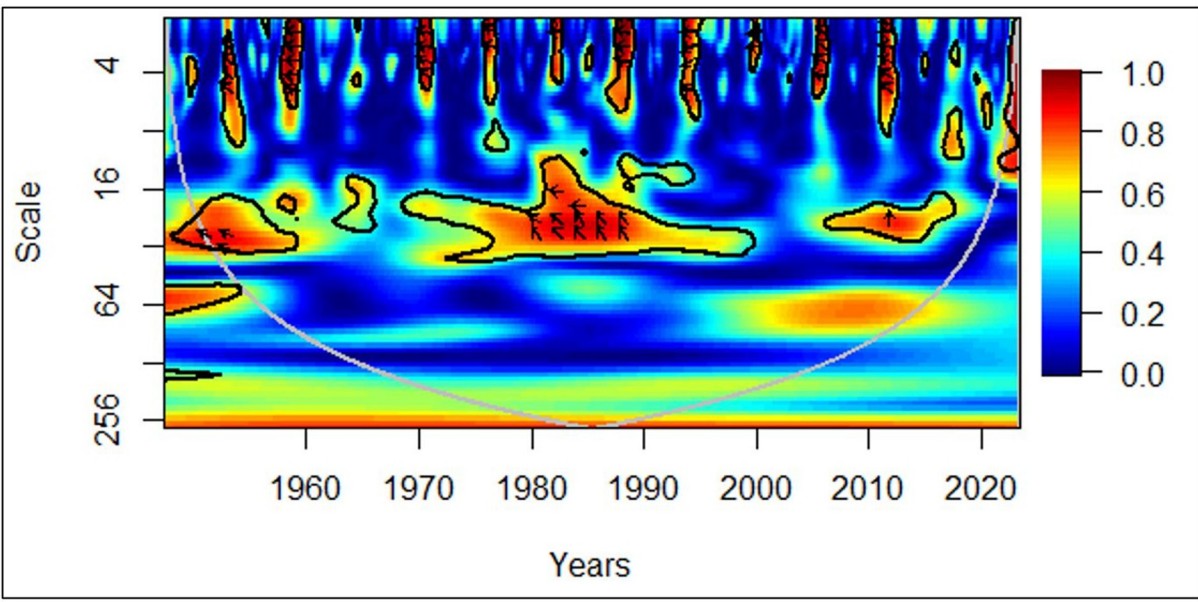

**Fig 10. South American region: GDP vs EPOP.** Source: Authors' illustration.

The results obtained from Granger causality test is provided in S3 Appendix. Moreover, S4 Appendix displays impulse response of the gdp and ddpop within itself and remaining variable. The diagonal panels in (a) and (d) shows the shock impact to future gdp and ddpop growth from own growth. In (a), in first and third phases ddpop have a standard deviation shock connected with its future values, the way it diminishes to zero. In (d), the gdp significantly responds to its own shock in the first phase and second phase. After checking the impulse response of the per capita GDP and elderly population, the stability condition is checked through root of companion matrix. The result of root of companion matrix eigenvalues confirms that the estimate is stable.

When considered the relationship between elderly population and economic growth in Asian countries separately, in general, there is reasonable confirmation that population aging and economic expansion Granger cause each other, with the exception of Indonesia, Iran, South Korea, Myanmar, the Philippines, Sri Lanka and Thailand. Results from Thailand, Indonesia and Sri Lanka are consistent with those of Dawson and Tiffin [55] and Chang, Chu [56]. They found that older people were neither the cause nor caused by economic growth. In other words, an increase in the elderly population has neither a positive nor a negative impact on economic growth.

For the European countries, bidirectional causality was not observed in any country, and non-causality was observed in 8 of the 15 countries in the data set (Belgium, Finland, France, Italy, Netherlands, Norway, Portugal, Turkey). In addition, unidirectional causality is observed in seven countries. For Greece and the United Kingdom, elderly population Granger caused economic growth, while five countries (Luxembourg, Austria, Denmark, Spain, and Sweden) are in the opposite direction with economic growth Granger causing elderly population. The decline in the number of countries with Granger bidirectional causality must have been due to several health factors, retirement planning, insurance planning, etc. Nonetheless, it is visible that nearly half of the sample countries have unidirectional Granger causality.

The causality observed in the current analysis indicates that Bolivia, Colombia, Guyana, Peru, and Puerto Rico have only one-way linear Granger causality from GDP per capita to elderly population. Costa Rica, Ecuador, and Honduras demonstrate a one-way causality from the elderly population to GDP per capita. No association of economic growth is shown with aging in Argentina, Bahamas, Belize, Brazil, Chile, Dominican Republic, Guatemala, Haiti, Latin America and the Caribbean, Mexico, Nicaragua, Panama, Paraguay, Suriname, Trinidad and Tobago, United States, and Uruguay.

For the African continent, Mauritania and Gabon have a unidirectional causality from elderly population to economic growth, while Benin, Botswana, Burundi, Madagascar, and Malawi have a unidirectional causality from economic growth to elderly population. Central African Republic and Senegal display a bidirectionality among the African countries. Rest of the African countries have no directionality between economic growth and elderly population.

When it comes to Oceania, only Fiji has a unidirectionality from economic growth to elderly population, whilst Australia and Papua New Guinea show a non-directionality. From this cross-country analysis, it is well evidenced that even among a region, nature of the relationship differs due to the institutional, geographic, and cultural differences across the countries and under varying macro-economic conditions.

## Conclusion

This research unveils the complex relationship between the economy and the elderly population. Even within a single region, subtle variations in the leading variable (i.e., economic growth and the elderly population) and the causality direction are visible over time. Even though existing literature has determined the causality between these two variables in dissimilar regions, those have only specified a general causality for the period under consideration. Since wavelet coherence was used as the methodology for this study, causality could be further analysed throughout the period along with its direction.

The major findings in this study are discussed in this paragraph. In a global perspective, it is palpable that predominantly the elderly population has led to economic growth in the positive direction towards the year 2000. Economic growth leading to elderly population inversely too could be seen sparsely. After 2000, no strong correlation or causality was visible between the two variables. In the African region, in the early 1980s, the elderly population negatively led the economic growth was visible in the medium-term. However, interestingly, this kind of behaviour has reversed from the 1990s till 2015 with the elderly population positively leading the economic growth. In the beginning, a positive causality is evident with the elderly population leading to economic growth in Asia. However, significant causality could not be seen afterwards. In Europe, the elderly population has inversely led to economic growth in 1960s. But over time, economic growth had inversely led to the elderly population of the region. Even though the elderly population leading to economic growth is visible, the dominant behaviour was the economic growth inversely leading the elderly population. A negative correlation between economic growth and the elderly population was evident in Oceania throughout the period under consideration. However, the leading variable changed from the elderly population to economic growth with time. Possibly, while the initial burden of elderly population on economy diminished, higher migration rates due to the economic growth must have resulted in economic growth inversely leading elderly population for Oceania. In the North American region, causality has been opposite to that of Oceania. Initially, economic growth was negatively leading the elderly population, but from the 21$^{st}$ century onwards, the elderly population have led the economic growth inversely. For the South American region, economic development dominated as the leading variable for the elderly population in the negative direction.

Results from the cross-country analysis prove how different policy implementations, and various macro-economic perspectives could impact the relationship between elderly population and economic growth among individual nations. Mixed results are obtained about the relationship with bi-directionality, non-directionality, and unidirectionality from both variables, depicting the complex nature of this relationship. Nonetheless, a cross-country analysis has led the research community to identify the impact of country specific differences when understanding the relationship among the elderly population and economic growth.

The finding drawn from this research could be helpful for global institutions and policy-making bodies, such as governments to learn from these patterns the real implications of the policy implementation regarding the elderly and the economy in the last six decades. From the lessons learnt from the past implementation, these authorities and actors can make much effective policies in the future to achieve their intended objectives.

It is strikingly clear from this research study that a much deeper investigation was facilitated in different regions regarding the nature between the economy and the elderly population. A possible future expansion of this study will be to examine the nature of the relationship and causality between the economy and the elderly population country-wise using the same dataset. Moreover, moderator variables, such as elderly care, that could have a significant impact on the relationship between the economy and the elderly population too could be explored. Apart from elderly care, considering possible influencers, such as quality of life, economic booms or recessions, disasters, and national and regional policies, could even add more value for a future study.

## Supporting information

**S1 Appendix. Descriptive statistics of the dataset.**
(DOCX)

**S2 Appendix. Data file.**
(XLSX)

**S3 Appendix. Cross-country analysis from Granger causality test.**
(DOCX)

**S4 Appendix. Impulse response of the per capita GDP and elderly population & root of companion matrix.**
(DOCX)

## Acknowledgments

The authors would like to thank Ms. Gayendri Karunarathne for proof-reading and editing this manuscript.

## Author Contributions

**Conceptualization:** Kethaka Galappaththi, Ruwan Jayathilaka, Lochana Rajamanthri.

**Data curation:** Kethaka Galappaththi, Thaveesha Jayawardhana, Sachini Anuththara, Thamasha Nimnadi, Ridhmi Karadanaarachchi.

**Formal analysis:** Kethaka Galappaththi, Ruwan Jayathilaka, Lochana Rajamanthri, Thaveesha Jayawardhana, Sachini Anuththara, Thamasha Nimnadi, Ridhmi Karadanaarachchi.

**Methodology:** Kethaka Galappaththi, Ruwan Jayathilaka, Lochana Rajamanthri, Thaveesha Jayawardhana, Sachini Anuththara, Thamasha Nimnadi, Ridhmi Karadanaarachchi.

**Resources:** Ruwan Jayathilaka.

**Software:** Kethaka Galappaththi, Lochana Rajamanthri.

**Supervision:** Kethaka Galappaththi, Ruwan Jayathilaka.

**Validation:** Kethaka Galappaththi, Ruwan Jayathilaka.

**Visualization:** Kethaka Galappaththi, Ruwan Jayathilaka, Lochana Rajamanthri, Thaveesha Jayawardhana, Sachini Anuththara, Thamasha Nimnadi, Ridhmi Karadanaarachchi.

**Writing – original draft:** Kethaka Galappaththi, Ruwan Jayathilaka, Lochana Rajamanthri, Thaveesha Jayawardhana, Sachini Anuththara, Thamasha Nimnadi, Ridhmi Karadanaarachchi.

**Writing – review & editing:** Ruwan Jayathilaka.

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
