## [Decision Letter · Decision Letter 0]

2 Oct 2022

PONE-D-22-22019Economy and Elderly Population, Complementary or Contradictory: A Cross-continental Wavelet Coherence StudyPLOS ONE

Dear Dr. Jayathilaka,

Thank you for submitting your manuscript to PLOS ONE. After careful consideration, we feel that it has merit but does not fully meet PLOS ONE’s publication criteria as it currently stands. Therefore, we invite you to submit a revised version of the manuscript that addresses the points raised during the review process.

We look forward to receiving your revised manuscript.

Kind regards,

Muhammad Ali, PhD

Academic Editor

PLOS ONE

Journal Requirements:

2. PLOS ONE does not copy edit accepted manuscripts (https://journals.plos.org/plosone/s/criteria-for-publication#loc-5). To that effect, please ensure that your submission is free of typos and grammatical errors.

3. We note you have included a table to which you do not refer in the text of your manuscript. Please ensure that you refer to Table 1 in your text; if accepted, production will need this reference to link the reader to the Table.

Additional Editor Comments:

Dear authors,

Thank you for submitting your paper to PloS One. As you can see, one of the two reviewers is suggesting straight rejection while the second one have suggested a major revision. After reading the comments in detail, I am using my discretion to give you another chance to respond to the reviewers therefore, I have decided that your paper can be resubmitted after major revisions in light of the comments given by the reviewers. Please submit your response in time.

All the best.

Reviewers' comments:

Reviewer's Responses to Questions

**Comments to the Author**

1. Is the manuscript technically sound, and do the data support the conclusions?

Reviewer #1: No

Reviewer #2: No

2. Has the statistical analysis been performed appropriately and rigorously? 

Reviewer #1: No

Reviewer #2: No

3. Have the authors made all data underlying the findings in their manuscript fully available?

Reviewer #1: No

Reviewer #2: Yes

4. Is the manuscript presented in an intelligible fashion and written in standard English?

Reviewer #1: No

Reviewer #2: Yes

5. Review Comments to the Author

Reviewer #1: The Abstract must report the aim of the study, the basic information on the sample (time span, countries analyzed), the empirical methodology used, the main findings, and the relevant policy implications.

Introduction and Literature Review should be split into two different sections.

The Introduction should highlight the relevance of the topic, the novelty of the results, the importance of policy implications, the sample’s choice, the methodology’s appropriateness, the data used, the contribution to the literature, and the limitations of the study.

The literature review is partial and incomplete, and some recent and relevant contributions should be cited and discussed: i.e., 10.1002/ijfe.2184; 10.1016/j.retrec.2021.101126; 10.1016/j.egyr.2021.03.005; 10.1016/j.jeca.2022.e00254.

Delete useless citations: Adebayo TS. New Insights into Export-growth Nexus: Wavelet and Causality Approaches. Asian Journal of Economics, Business and Accounting. 2020;15(2):32-44. doi: 10.9734/ajeba/2020/v15i230212; Sunday Adebayo T. Dynamic Relationship between Oil Price and Inflation in Oil Exporting Economy: Empirical Evidence from Wavelet Coherence Technique. Energy Economics Letters. 2020;7(1):12-22. doi: 10.18488/journal.82.2020.71.12.22. Alola AA, Kirikkaleli D. The nexus of environmental quality with renewable consumption, immigration, and healthcare in the US: wavelet and gradual-shift causality approaches. Environmental Science and Pollution Research. 2019;26(34):35208-17. doi: 10.1007/s11356-019-06522-y. PubMed PMID: 31696425. Kalmaz DB, Kirikkaleli D. Modeling CO 2 emissions in an emerging market: empirical finding from ARDL-based bounds and wavelet coherence approaches. Environmental

Science and Pollution Research. 2019;26(5):5210-20. doi: 10.1007/s11356-018-3920-z. PubMed PMID: 30604366.

The theoretical framework should be discussed more in detail.

The estimated model must be justified in light of the literature on this specific topic.

Descriptive statistics are absent.

Diagnostic tests are absent.

Robustness checks are absent.

The results should be discussed more in detail.

Comparisons with previous studies are absent.

Conclusions are too short.

Policy implications are weak.

Further research should be indicated.

Limitations of the study are not provided.

Proofreading by a native speaker is required.

The editing does not follow the journal’s guidelines.

The originality value of the study is limited.

This is a basic econometric exercise without a clear innovative intuition.

How does the paper enrich the knowledge of the scientific community?

Reviewer #2: This study investigates the causal relationship between the economy and the elderly population globally as well as continent-wise to investigate the differences between several regions simultaneously using time series data. The dependent variable in this study is GDP (per capita growth rate) while the targeted population aged above 65 as a percentage of the total was considered the elderly population as the independent variable. The study utilizes the panel dataset published by the World Bank for a period of six decades from 1961 to 2020 covering 84 countries. Wavelet coherence was the methodology used for the study since it was suitable to present causality as well as the causal direction between two variables for different sections during the six decades.

One of the major concerns about the study is concerning the institutional, geographic, and cultural differences across the countries which the authors did not incorporate. It is important to argue about cross-country differences and control such variations in the model before establishing the causality at the intersection of the mentioned idea. There is a possibility that some of the countries used in the data might have a higher elderly adult population due to better healthcare infrastructure or other institutional differences which might affect GDP differently than others.

Additionally, I am unable to understand why the author wants to revisit the already established relationship because there is an enormous amount of research already available with the cross-country analysis where the researchers have already documented this phenomenon with the help of longitudinal data.

There are some flaws in the write-up of this paper, the author is unable to provide specific references to the relationship between GDP and the elderly population. In the introduction, the author argued that previous studies revealed that there is a negative relationship between the elderly population and GDP but this study concludes that there is a positive relationship, now this is a serious claim that demands a detailed explanation of why it has a positive relationship. In fact this does not go well with the established economic theory that why there is a positive relationship between GDP and elderly adult population. I did not see any justification in this regard either and here come the institutional and other differences which the study lacks. Although the wavelet transform (DWT) is a powerful tool for signal and image processing, but it is affected by sensitivity, poor directionality, and lack of phase information. The casual relationship is not discussed further in the paper as discussed in the abstract.

6. PLOS authors have the option to publish the peer review history of their article (what does this mean?). If published, this will include your full peer review and any attached files.

Reviewer #1: No

Reviewer #2: **Yes: **SYED HASSAN RAZA

---

## [Author Response · Author response to Decision Letter 0]

10 Nov 2022

Point by point response to reviewers

Dear Reviewers,

We would like to express our profound appreciation to the reviewers for the valuable comments and suggestions made on our manuscript which were very helpful in revising and improving it. Please note that the line numbers referred in this document is aligned with the revised manuscript which has track changes.

Reviewer 1 comment 1: The Abstract must report the aim of the study, the basic information on the sample (time span, countries analyzed), the empirical methodology used, the main findings, and the relevant policy implications.

Response: Thank you very much and the revised version has strengthened the aim. The information on time span and countries analysed, and the empirical methodology used in the study are stated in the abstract from line 32 to 35.

“A panel dataset published by the World Bank for a period of six decades from 1961 to 2020 covering 84 countries was used as data for the analysis. Wavelet coherence was the methodology used for the study since it was considered suitable to present causality as well as the causal direction between two variables for different sections during the six decades.” 

Findings and policy implications are provided in the revised manuscript from line 37 to 41.

“The findings of the study reveal that the causality and its direction have been changing over time for most continents. Negative correlations with the leading variable interchanging with time are evident for the majority of the regions. Nevertheless, results indicate that in global perspective, predominantly elderly population leads the economic growth with positive correlation.”

Reviewer 1 comment 2: Introduction and Literature Review should be split into two different sections. The Introduction should highlight the relevance of the topic, the novelty of the results, the importance of policy implications, the sample’s choice, the methodology’s appropriateness, the data used, the contribution to the literature, and the limitations of the study.

Response: Thank you for the detailed comment. The Introduction and literature review have been split into two different sections, then further revised and extended. 

Introduction contains in revised manuscript from line 47 to 89 and literature review contains in revised manuscript from line 90 to 131.

Specifically, the introduction has been revised to highlight the relevance of the topic, the novelty of the results, the importance of policy implications, the sample’s choice, the methodology’s appropriateness, the data used, the contribution to the literature, and the limitations of the study, as given in revised manuscript from Line 53 to 58; line 73 to 84

“…According to the World Development Indicators of the World Bank [1], 6% of the elderly population in the 1980s has grown to 7% in the 1990s, and 10% in 2021. In general, an increase in life expectancy with medical advancements and low fertility rates have been identified as the root causes for the rapid growth of the elderly population. However, it is imperative to reflect on the consequences of the situation and to set national policies accordingly…”

"…The significance of this study is evident in many different aspects. Firstly, the novelty of the research is due to the initial usage of the wavelet coherence approach, as the methodology adopted to investigate the nature of the relationship and causality between the economy and the elderly population. This approach allows ascertaining the short-term and medium-term changes that occurred concerning the direction of the relationship throughout the stipulated period of the study, which could not be drawn by any previous study. Secondly, the study considers over 80 countries worldwide representing all habitable continents for 60 years. Even though region-wise literature is available on this topic, global studies for decades have not been conducted yet. Thirdly, the ability of this research to distinguish variations within the stipulated period for each region would be helpful. This will assist to ensure the effectiveness of previous policy implementations related to the economy and the elderly population and its learnings could be used for future decision making…”

Reviewer 1 comment 3: The literature review is partial and incomplete, and some recent and relevant contributions should be cited and discussed: i.e., 10.1002/ijfe.2184; 10.1016/j.retrec.2021.101126; 10.1016/j.egyr.2021.03.005; 10.1016/j.jeca.2022.e00254.

Delete useless citations: Adebayo TS. New Insights into Export-growth Nexus: Wavelet and Causality Approaches. Asian Journal of Economics, Business and Accounting. 2020;15(2):32-44. doi: 10.9734/ajeba/2020/v15i230212; Sunday Adebayo T. Dynamic Relationship between Oil Price and Inflation in Oil Exporting Economy: Empirical Evidence from Wavelet Coherence Technique. Energy Economics Letters. 2020;7(1):12-22. doi: 10.18488/journal.82.2020.71.12.22. Alola AA, Kirikkaleli D. The nexus of environmental quality with renewable consumption, immigration, and healthcare in the US: wavelet and gradual-shift causality approaches. Environmental Science and Pollution Research. 2019;26(34):35208-17. doi: 10.1007/s11356-019-06522-y. PubMed PMID: 31696425. Kalmaz DB, Kirikkaleli D. Modeling CO 2 emissions in an emerging market: empirical finding from ARDL-based bounds and wavelet coherence approaches. Environmental

Science and Pollution Research. 2019;26(5):5210-20. doi: 10.1007/s11356-018-3920-z. PubMed PMID: 30604366.

Response: Noted with thanks! we revised our citations taking into account the references suggested by this Reviewer and adding new and appropriate past literature.

Citations recommended to be deleted were deleted and newly added citations are available in revised manuscript in line 169.

“…Several research studies have further contributed to the development of the model, including the wavelet transformation of the two time series and the wavelet coherence at difference phases [34-37].”

New references are available in References section of the revised manuscript from line 574 to 583.

“34. Magazzino C, Giolli L. The relationship among railway networks, energy consumption, and real added value in Italy. Evidence form ARDL and Wavelet analysis. Research in Transportation Economics. 2021;90:101126. doi: https://doi.org/10.1016/j.retrec.2021.101126.

35. Magazzino C, Mutascu M, Mele M, Sarkodie SA. Energy consumption and economic growth in Italy: A wavelet analysis. Energy Reports. 2021;7:1520-8. doi: https://doi.org/10.1016/j.egyr.2021.03.005.

36. Magazzino C, Mutascu MI. The Italian fiscal sustainability in a long-run perspective. The Journal of Economic Asymmetries. 2022;26:e00254. doi: https://doi.org/10.1016/j.jeca.2022.e00254. …”

Reviewer 1 comment 4: The theoretical framework should be discussed more in detail

.

Response: Thank you for pointing out this point for improvement. A theoretical framework in discussion was missing in the original version of the paper. To fill this gap, in the revised manuscript, we added the theoretical framework in figure 1 in page 11 and explained from line 214 to 217

“...The theoretical framework clearly depicts the study aim, to identify/understand the nature between two endogenous variables, elderly population and economic growth, operationalised by the population aged 65 and above as a percentage from the total population and annual GDP per capita growth rate, respectively...”

Reviewer 1 comment 5: The estimated model must be justified in light of the literature on this specific topic.

Response: Noted with thanks! Justification on the estimated model was provided through the literature review in the revised manuscript from line 127 to 131

“…Although sufficient literature is available about the relationship between the elderly population and economic growth, these lack the insights from different periods of time as it is a known fact that the relationship would change due to short-term and medium-term policy implementations and other macro-economic conditions. Thus, the model derived in this study fills the literature gap that is yet to be filled…”

Reviewer 1 comment 6: Descriptive statistics are absent.

Response: Thank you very much for the comment! Descriptive statistics of the dataset has been added to the appendix S1 and it was referred in the revised manuscript in the line 162

“Descriptive statistics of the dataset is provided in the Appendix S1”

Reviewer 1 comment 7: Diagnostic tests are absent. Robustness checks are absent. 

Response: Well noted with thanks! Impulse response and Roots of the companion matrix tests were conducted and included in the appendix S4 and referred in the revised manuscript in the lines from 203 to 204 and 389 to 397.

“Furthermore, as diagnostics tests, impulse response of the per capita GDP and elderly population for all countries was conducted along with root of companion matrix.”

“The results obtained from Granger causality test is provided in S3 Appendix. Moreover, Appendix S4 displays impulse response of the gdp and ddpop within itself and remaining variable. The diagonal panels in (a) and (d) shows the shock impact to future gdp and ddpop growth from own growth. In (a), in first and third phases ddpop have a standard deviation shock connected with its future values, the way it diminishes to zero. In (d), the gdp significantly responds to its own shock in the first phase and second phase. After checking the impulse response of the per capita GDP and elderly population, the stability condition is checked through root of companion matrix. The result of root of companion matrix eigenvalues confirms that the estimate is stable.”

Reviewer 1 comment 8: The results should be discussed more in detail.

Response: Thanks, results and discussion section was elaborated along with a cross-country analysis in the revised manuscript from line 384 to line 432

“In order to incorporate the institutional, geographic, and cultural differences across the countries, a cross-country analysis was conducted. The reason is because there is a possibility that some countries included in the dataset might have a higher elderly adult population due to better healthcare infrastructure or other institutional differences which might affect GDP differently than others.

The results obtained from Granger causality test is provided in S3 Appendix. Moreover, Appendix S4 displays impulse response of the gdp and ddpop within itself and remaining variable. The diagonal panels in (a) and (d) shows the shock impact to future gdp and ddpop growth from own growth. In (a), in first and third phases ddpop have a standard deviation shock connected with its future values, the way it diminishes to zero. In (d), the gdp significantly responds to its own shock in the first phase and second phase. After checking the impulse response of the per capita GDP and elderly population, the stability condition is checked through root of companion matrix. The result of root of companion matrix eigenvalues confirms that the estimate is stable.

When considered the relationship between elderly population and economic growth in Asian countries separately, in general, there is reasonable confirmation that population aging and economic expansion Granger cause each other, with the exception of Indonesia, Iran, South Korea, Myanmar, the Philippines, Sri Lanka and Thailand. Results from Thailand, Indonesia and Sri Lanka are consistent with those of Dawson and Tiffin [55] and Chang, Chu [56]. They found that older people were neither the cause nor caused by economic growth. In other words, an increase in the elderly population has neither a positive nor a negative impact on economic growth.

For the European countries, bidirectional causality was not observed in any country, and non-causality was observed in 8 of the 15 countries in the data set (Belgium, Finland, France, Italy, Netherlands, Norway, Portugal, Turkey). In addition, unidirectional causality is observed in seven countries. For Greece and the United Kingdom, elderly population Granger caused economic growth, while five countries (Luxembourg, Austria, Denmark, Spain, and Sweden) are in the opposite direction with economic growth Granger causing elderly population. The decline in the number of countries with Granger bidirectional causality must have been due to several health factors, retirement planning, insurance planning, etc. Nonetheless, it is visible that nearly half of the sample countries have unidirectional Granger causality.

The causality observed in the current analysis indicates that Bolivia, Colombia, Guyana, Peru, and Puerto Rico have only one-way linear Granger causality from GDP per capita to elderly population. Costa Rica, Ecuador, and Honduras demonstrate a one-way causality from the elderly population to GDP per capita. No association of economic growth is shown with aging in Argentina, Bahamas, Belize, Brazil, Chile, Dominican Republic, Guatemala, Haiti, Latin America and the Caribbean, Mexico, Nicaragua, Panama, Paraguay, Suriname, Trinidad and Tobago, United States, and Uruguay.

For the African continent, Mauritania and Gabon have a unidirectional causality from elderly population to economic growth, while Benin, Botswana, Burundi, Madagascar, and Malawi have a unidirectional causality from economic growth to elderly population. Central African Republic and Senegal display a bidirectionality among the African countries. Rest of the African countries have no directionality between economic growth and elderly population.

When it comes to Oceania, only Fiji has a unidirectionality from economic growth to elderly population, whilst Australia and Papua New Guinea show a non-directionality. From this cross-country analysis, it is well evidenced that even among a region, nature of the relationship differs due to the institutional, geographic, and cultural differences across the countries and under varying macro-economic conditions.”

Reviewer 1 comment 9: Comparisons with previous studies are absent.

Response: Many thanks for the comment. The comparison with previous studies was conducted and mentioned in revised manuscript from line 92 to line 115

“Opinion of research community is divided regarding the relationship between economic growth and the elderly population. A substantial research studies conclude a negative relationship [7-12] while a considerable amount of existing literature suggest a positive relationship [13-16]. Despite the fact that the research community has shown interest about the nature of the relationship and causality between the elderly population and economic growth, the importance of identifying its relationship is unchallenged. Even with the ample body of literature available in this field, the change in the nature of the relationship between the economy and the elderly population is yet to be investigated in the global scale for a prolonged period. At present, a few regional studies have been conducted across regions [17, 18] since individual countries [19-22] had been the main focus in numerous research studies. The existing literature had been using different methodologies such as regression [23, 24] and granger causality [25-27] to identify the effects one variable has on the other and the direction of those two variables. However, so far, according to available information and researcher’s knowledge, no study has ever been conducted to ascertain the relationship between the economy and the elderly population across habitable continents for several decades, with the emphasis given for changes to the nature of the relationship along that time period. Yet, it is of utmost importance to look into such changes concerning the relationship. It is because national, regional, and international institutions have implemented various policies from time to time, which would have affected the nature of this relationship. Thus, a clear literature gap is visible which is to be filled from the findings of this research study.”

Reviewer 1 comment 10: Conclusions are too short. Policy implications are weak. Further research should be indicated. Limitations of the study are not provided.

Response Duly noted with thanks! Conclusion was strengthened with further elaboration on policy implications, further research and limitations of the study as provided in the revised manuscript from line 433 to 485 Also, the word count for conclusion section was increased from 572 to 651 words.

Reviewer 1 comment 11: Proofreading by a native speaker is required.

Response: The paper has been revised thoroughly and in-depth proofreading check has been performed by a linguistic professional. We can confirm that the revised manuscript is free of any language errors.

Reviewer 1 comment 12: The editing does not follow the journal’s guidelines.

Response: We thank the Reviewer for this check. We revised the manuscript to comply with PLOS ONE Submission Guidelines.

(https://journals.plos.org/plosone/s/submission-guidelines)

Reviewer 1 comment 13: The originality value of the study is limited.

Response: Thanks for the comment. Originality and significance of the study come in different aspects. Discussion on the originality was emphasised. Revised manuscript better highlights the contribution of the work, in particular the Introduction from line 73 to 84

“…The significance of this study is evident in many different aspects. Firstly, the novelty of the research is due to the initial usage of the wavelet coherence approach, as the methodology adopted to investigate the nature of the relationship and causality between the economy and the elderly population. This approach allows ascertaining the short-term and medium-term changes that occurred concerning the direction of the relationship throughout the stipulated period of the study, which could not be drawn by any previous study. Secondly, the study considers over 80 countries worldwide representing all habitable continents for 60 years. Even though region-wise literature is available on this topic, global studies for decades have not been conducted yet. Thirdly, the ability of this research to distinguish variations within the stipulated period for each region would be helpful. This will assist to ensure the effectiveness of previous policy implementations related to the economy and the elderly population and its learnings could be used for future decision making…”

Reviewer 1 comment 14: This is a basic econometric exercise without a clear innovative intuition.

Response: Whilst thanking for the comment, we would like to bring out the consideration that wavelet analysis is an innovative approach for the elderly population and economic growth relationship with many insights extracted in terms of causality as well as directionality. Moreover, as per the author’s knowledge, the innovative technique, wavelet approach has not been applied for elderly population and economic growth as given in revised manuscript from line 128 to 131.

“…these lack the insights from different periods of time as it is a known fact that the relationship would change due to short-term and medium-term policy implementations and other macro-economic conditions. Thus, the model derived in this study fills the literature gap that is yet to be filled…”

Reviewer 1 comment 15: How does the paper enrich the knowledge of the scientific community?

Response: Thanks for the consideration! The paper enriches the knowledge of the scientific community with the introduction of the wavelet approach on the elderly population and economic growth to investigate the nature of the relationship and causality between the economy and the elderly population. This approach allows ascertaining the short-term and medium-term changes that occurred concerning the direction of the relationship throughout the stipulated period of the study (recent six decades), which could not be drawn by any previous study. This fact was emphasised in revised manuscript from line 74 to 82.

“…Firstly, the novelty of the research is due to the initial usage of the wavelet coherence approach, as the methodology adopted to investigate the nature of the relationship and causality between the economy and the elderly population. This approach allows ascertaining the short-term and medium-term changes that occurred concerning the direction of the relationship throughout the stipulated period of the study, which could not be drawn by any previous study. Secondly, the study considers over 80 countries worldwide representing all habitable continents for 60 years. Even though region-wise literature is available on this topic, global studies for decades have not been conducted yet. Thirdly, the ability of this research to distinguish variations within the stipulated period for each region would be helpful…”

Reviewer 2 comment 1: One of the major concerns about the study is concerning the institutional, geographic, and cultural differences across the countries which the authors did not incorporate. It is important to argue about cross-country differences and control such variations in the model before establishing the causality at the intersection of the mentioned idea. There is a possibility that some of the countries used in the data might have a higher elderly adult population due to better healthcare infrastructure or other institutional differences which might affect GDP differently than others.

Response: Thank you for the valuable comment! We agree with you the institutional, geographic, and cultural differences across the countries and the importance to incorporate cross-country differences and control such variations in the model before establishing the causality at the intersection.

Taking the comment into consideration, cross-country analysis was conducted in order to understand the differences of the nature of relationship among elderly population and economy for individual countries and mentioned in the revised manuscript from line 384 to 432

“…In order to incorporate the institutional, geographic, and cultural differences across the countries, a cross-country analysis was conducted. The reason is because there is a possibility that some countries included in the dataset might have a higher elderly adult population due to better healthcare infrastructure or other institutional differences which might affect GDP differently than others.

The results obtained from Granger causality test is provided in S3 Appendix. Moreover, Appendix S4 displays impulse response of the gdp and ddpop within itself and remaining variable. The diagonal panels in (a) and (d) shows the shock impact to future gdp and ddpop growth from own growth. In (a), in first and third phases ddpop have a standard deviation shock connected with its future values, the way it diminishes to zero. In (d), the gdp significantly responds to its own shock in the first phase and second phase. After checking the impulse response of the per capita GDP and elderly population, the stability condition is checked through root of companion matrix. The result of root of companion matrix eigenvalues confirms that the estimate is stable.

When considered the relationship between elderly population and economic growth in Asian countries separately, in general, there is reasonable confirmation that population aging and economic expansion Granger cause each other, with the exception of Indonesia, Iran, South Korea, Myanmar, the Philippines, Sri Lanka and Thailand. Results from Thailand, Indonesia and Sri Lanka are consistent with those of Dawson and Tiffin [55] and Chang, Chu [56]. They found that older people were neither the cause nor caused by economic growth. In other words, an increase in the elderly population has neither a positive nor a negative impact on economic growth.

For the European countries, bidirectional causality was not observed in any country, and non-causality was observed in 8 of the 15 countries in the data set (Belgium, Finland, France, Italy, Netherlands, Norway, Portugal, Turkey). In addition, unidirectional causality is observed in seven countries. For Greece and the United Kingdom, elderly population Granger caused economic growth, while five countries (Luxembourg, Austria, Denmark, Spain, and Sweden) are in the opposite direction with economic growth Granger causing elderly population. The decline in the number of countries with Granger bidirectional causality must have been due to several health factors, retirement planning, insurance planning, etc. Nonetheless, it is visible that nearly half of the sample countries have unidirectional Granger causality.

The causality observed in the current analysis indicates that Bolivia, Colombia, Guyana, Peru, and Puerto Rico have only one-way linear Granger causality from GDP per capita to elderly population. Costa Rica, Ecuador, and Honduras demonstrate a one-way causality from the elderly population to GDP per capita. No association of economic growth is shown with aging in Argentina, Bahamas, Belize, Brazil, Chile, Dominican Republic, Guatemala, Haiti, Latin America and the Caribbean, Mexico, Nicaragua, Panama, Paraguay, Suriname, Trinidad and Tobago, United States, and Uruguay.

For the African continent, Mauritania and Gabon have a unidirectional causality from elderly population to economic growth, while Benin, Botswana, Burundi, Madagascar, and Malawi have a unidirectional causality from economic growth to elderly population. Central African Republic and Senegal display a bidirectionality among the African countries. Rest of the African countries have no directionality between economic growth and elderly population.

When it comes to Oceania, only Fiji has a unidirectionality from economic growth to elderly population, whilst Australia and Papua New Guinea show a non-directionality. From this cross-country analysis, it is well evidenced that even among a region, nature of the relationship differs due to the institutional, geographic, and cultural differences across the countries and under varying macro-economic conditions…”

Reviewer 2 comment 2: I am unable to understand why the author wants to revisit the already established relationship because there is an enormous amount of research already available with the cross-country analysis where the researchers have already documented this phenomenon with the help of longitudinal data.

Response: While we also agree that relationship between GDP and the elderly population has already been investigated among different countries, this study is significant compared with the rest due to following reasons,

It contains a continent-wise study covering 84 countries throughout the most recent 60 years. As per the authors’ knowledge, no research has been conducted containing this kind of a large panel dataset. 

Use of the methodology, wavelet analysis, has enabled the study to discuss important insights year-wise in visual manner. Also, it showed not only the direction of the relationship, but also the causality between two variables. Wavelet analysis has not been used to test the nature of two variables.

Additionally, cross-country analysis was conducted for 84 countries to understand individual nation’s behaviour due to policy and macro-economic differences among them.

These were included in the revised manuscript from line 67 to 84.

“…This research was designed as a continent-wide study to simultaneously investigate the differences between several regions. Furthermore, a cross-country analysis was conducted using Granger causality to understand the relationship between the economy and the elderly population among individual countries. It is because countries have implemented diverse policies and varying macro-economic conditions exist, which would impact on the relationship of the two variables within a given country. The significance of this study is evident in many different aspects. Firstly, the novelty of the research is due to the initial usage of the wavelet coherence approach, as the methodology adopted to investigate the nature of the relationship and causality between the economy and the elderly population. This approach allows ascertaining the short-term and medium-term changes that occurred concerning the direction of the relationship throughout the stipulated period of the study, which could not be drawn by any previous study. Secondly, the study considers over 80 countries worldwide representing all habitable continents for 60 years. Even though region-wise literature is available on this topic, global studies for decades have not been conducted yet. Thirdly, the ability of this research to distinguish variations within the stipulated period for each region would be helpful. This will assist to ensure the effectiveness of previous policy implementations related to the economy and the elderly population, and its learnings could be used for future decision making…”

Reviewer 2 comment 3: There are some flaws in the write-up of this paper, the author is unable to provide specific references to the relationship between GDP and the elderly population. In the introduction, the author argued that previous studies revealed that there is a negative relationship between the elderly population and GDP but this study concludes that there is a positive relationship, now this is a serious claim that demands a detailed explanation of why it has a positive relationship. In fact this does not go well with the established economic theory that why there is a positive relationship between GDP and elderly adult population. I did not see any justification in this regard either and here come the institutional and other differences which the study lacks.

Response: Thank you for the comments!

We have discussed for both sides. In the revised version, it was elaborated. Specific references to the relationship between GDP and the elderly population were provided in a clearer manner with literature on both directions in the revised manuscript from line 92 to 95

“Opinion of research community is divided regarding the relationship between economic growth and the elderly population. A substantial research studies conclude a negative relationship [7-12] while a considerable amount of existing literature suggest a positive relationship [13-16].”

---

## [Decision Letter · Decision Letter 1]

22 Nov 2022

Economy and Elderly Population, Complementary or Contradictory: A Cross-continental Wavelet Coherence and Cross-country Granger Causality Study

PONE-D-22-22019R1

Dear Dr. Jayathilaka,

We’re pleased to inform you that your manuscript has been judged scientifically suitable for publication and will be formally accepted for publication once it meets all outstanding technical requirements.

Kind regards,

Muhammad Ali, PhD

Academic Editor

PLOS ONE

Additional Editor Comments:

Dear authors

I have checked and also received the confirmation from the reviewers that you have incorporated all the comments. I think your paper is in good shape and I am accepting it for publication. Congratulations. The publication team will be in touch with you shortly.

Warm regards

Ali

Reviewers' comments:

Reviewer's Responses to Questions

**Comments to the Author**

1. If the authors have adequately addressed your comments raised in a previous round of review and you feel that this manuscript is now acceptable for publication, you may indicate that here to bypass the “Comments to the Author” section, enter your conflict of interest statement in the “Confidential to Editor” section, and submit your "Accept" recommendation.

Reviewer #1: All comments have been addressed

2. Is the manuscript technically sound, and do the data support the conclusions?

Reviewer #1: Yes

3. Has the statistical analysis been performed appropriately and rigorously? 

Reviewer #1: Yes

4. Have the authors made all data underlying the findings in their manuscript fully available?

Reviewer #1: Yes

5. Is the manuscript presented in an intelligible fashion and written in standard English?

Reviewer #1: Yes

6. Review Comments to the Author

Reviewer #1: ----------------------------------------------------------------------------------------------------------------------------------------------------

7. PLOS authors have the option to publish the peer review history of their article (what does this mean?). If published, this will include your full peer review and any attached files.

Reviewer #1: No

---

## [Editor Report · Acceptance letter]

15 Dec 2022

PONE-D-22-22019R1 

Economy and Elderly Population, Complementary or Contradictory: A Cross-continental Wavelet Coherence and Cross-country Granger Causality Study 

Dear Dr. Jayathilaka:

I'm pleased to inform you that your manuscript has been deemed suitable for publication in PLOS ONE. Congratulations! Your manuscript is now with our production department. 

Kind regards, 

on behalf of

Dr. Muhammad Ali 

Academic Editor

PLOS ONE